# Model-checking ecological state-transition graphs

**Colin Thomas**[1,2]*, **Maximilien Cosme**[2], **Cédric Gaucherel**[2], **Franck Pommereau**[1]*

**1** IBISC, Univ. Évry, Univ. Paris-Saclay, 91020 Évry-Courcouronne, France, **2** AMAP, Univ. Montpellier, INRAE, CIRAD, CNRS, IRD, Montpellier, France

\* colin.thomas@univ-evry.fr (CT); franck.pommereau@univ-evry.fr (FP)

**Data Availability Statement:** All relevant data are within the manuscript and its Supporting information files.

**Funding:** CG was partly funded by the Eranet LEAP-Agri project SESASA (n° 01DG18020) https://

## Abstract

Model-checking is a methodology developed in computer science to automatically assess the dynamics of discrete systems, by checking if a system modelled as a state-transition graph satisfies a dynamical property written as a temporal logic formula. The dynamics of ecosystems have been drawn as state-transition graphs for more than a century, ranging from state-and-transition models to assembly graphs. Model-checking can provide insights into both empirical data and theoretical models, as long as they sum up into state-transition graphs. While model-checking proved to be a valuable tool in systems biology, it remains largely underused in ecology apart from precursory applications.

This article proposes to address this situation, through an inventory of existing ecological STGs and an accessible presentation of the model-checking methodology. This overview is illustrated by the application of model-checking to assess the dynamics of a vegetation pathways model. We select management scenarios by model-checking Computation Tree Logic formulas representing management goals and built from a proposed catalogue of patterns. In discussion, we sketch bridges between existing studies in ecology and available model-checking frameworks. In addition to the automated analysis of ecological state-transition graphs, we believe that defining ecological concepts with temporal logics could help clarify and compare them.

## Author summary

Ecologists have drawn state-transition graphs representing the dynamics of ecosystems for more than a century. Model-checking is an automated method for the analysis of such graphs developed in computer science and acknowledged by a Turing award in 2007. Ecologists appear to be mostly unaware of model-checking apart from precursory applications, despite its successes in systems biology.

We propose to address this situation, through an inventory of existing ecological state-transition graphs and an accessible presentation of the model-checking methodology. We exemplify the insights provided by model-checking by applying it to a vegetation pathways model in order to assess management policies aiming to tackle savanna encroachment. We provide a catalogue of patterns to help ecologists with the difficulty of formally

leap-agri.com/?page_id=293. The funders had no role in study design, data collection and analysis, decision to publish, or preparation of the manuscript.

**Competing interests:** The authors have declared that no competing interests exist.

expressing dynamical properties. Lastly, we sketch bridges between existing studies in ecology and available model-checking frameworks.

Model-checking can be applied to both empirical data and theoretical models, as long as they sum up into state-transition graphs. It provides automated and accurate answers to complex questions that could barely be analysed through human examination, if not impossible to answer this way. In addition, we believe that formally defining ecological concepts within the model-checking framework could help clarify and compare them.

This is a *PLOS Computational Biology* Methods paper.

## Introduction

A *state-transition graph (STG)* describes the behaviour of a dynamical system, for example an ecosystem, as a graph whose nodes are the discrete states of the system and whose edges represent the transitions between those states. Ecologists have drawn STGs for more than a century, one of the earliest and best-known examples being the vegetation successions described by Clements [1]. Yet, ecology and environmental sciences appear to remain largely unaware of the *model-checking* methodology [2] developed in computer science to investigate the dynamics of a system represented as an STG. This paper aims to promote the model-checking of ecological STGs.

In ecology, STGs are typically used to represent *community pathways*, i.e. changes in the set of species or populations of an ecosystem through time. For example the successions of plant communities in boreal forests [3], or the assembly of protist communities in laboratory experiments [4]. Such STGs are mostly drawn from observations, hence their relatively small size (a few dozens of states at most). Most of the time, STGs are perceived as graphical representations of the knowledge about the dynamics of the studied system rather than as actual data.

For example, STGs have been used since the '90s as a tool for rangeland management and ecosystem conservation under the concept of *state-and-transition models (STMs)* [5, 6]. Theoretical studies also emphasise the relevance of STGs to investigate community assembly [7, 8]. Both research fields recently mentioned an interest in tools providing dynamical analysis of STGs [8, 9].

In computer science, STGs model the executions of automated systems. Computer scientists design automated tools called *model-checkers* to ensure the absence of bugs during software executions [2]. Model-checkers verify whether the pathways within an STG satisfy a given property, for example that a desired state remains reachable or that harmful behaviour is always avoided. Given a system description that can be computed into an STG and a dynamical property written as a *temporal logic formula*, a model-checker outputs whether the STG satisfies the property or not. Model-checking is an active field of research acknowledged by a Turing award in 2007, encompassing numerous concepts and resulting in a wide variety of implemented tools [10, 11].

In systems biology, STGs are outputted by models of reaction networks or regulatory networks [12]. Model-checking is extensively used to analyse those models [13, 14], proving its suitability for the study of biological systems. For example, model-checking helped validate models of nutritional stress response of *Escherichia coli* [15], T-helper cell reprogramming

[16], mammalian cell cycle [17] or BRAF inhibition pathways in two different cancers [18]. Yet, model-checking methodology appears to remain unknown to most ecologists apart from precursory applications [19, 20]. Ecology encompasses a wide variety of STGs but their analysis is often restricted to visual examination.

This article proposes to address this situation by helping the ecologists handling STGs to get acquainted with the model-checking methodology. First, it provides an inventory of existing ecological STGs and a didactic presentation of model-checking. This overview is illustrated by the application of model-checking to a model of the Borana vegetation pathways based on STM literature [21–24]. The model-checking methodology can be implemented in a wide variety of ways, bridges between existing studies in ecology and available model-checking frameworks are sketched in Discussion.

## 1 Materials and methods

### 1.1 State-transition graphs (STGs)

A *state-transition graph (STG)* [25] represents the dynamics of a system as a *graph G = (S, E)*, where *S* is a set of *nodes* (the discrete states of the system) and $E \subseteq S \times S$ is a set of *directed edges* (the transitions enabling to move from one state to another). The fact that $(s, s') \in E$ is often noted $s \rightarrow s'$.

The transitions may be labelled by their driving event or process (taken from a set of labels *L*), resulting in $E \subseteq S \times L \times S$, the fact that $(s, l, s') \in E$ is then noted $s \xrightarrow{l} s'$. An STG is said *deterministic* if every state has at most one outgoing transition, and *non-deterministic* otherwise. Every state of a deterministic STG is the start of a single pathway, thus the behaviour of the system is also deterministic.

In ecology, the state $s \in S$ of an ecosystem is often discretely abstracted by its *community* (i.e. restricted to its set of species or populations). Subsequently, STGs are found in a broad variety of studies focusing on the dynamics of ecological communities, historically called community succession for plants and community assembly for animals [26, 27].

Graphs are widespread in ecology, but STGs must be discriminated from *interaction networks* such as the iconic trophic networks [28, 29]. Indeed, the former grasp the temporal behaviour of an ecological system, while the latter grasp the processes taking place between its components. A node (resp. an edge) of an STG is a temporal stage (resp. an event, i.e. a state transition) of the system dynamics. Whereas a node (resp. an edge) of an interaction network is a component (resp. a flux) of the system. The model-checking methodology presented in this paper deals with the temporal changes of a discrete-event system and thus is designed to analyse STGs specifically.

**1.1.1 State-transition graphs in ecology.** The dynamics of ecological systems have been described as states and transitions for more than a century. For example, Clements [1] used STGs to represent ecological successions [30] between vegetation states called "seral stages". Since then, STGs have regularly been used under various names in ecology, from "behaviour graphs" [31], to "kinematic graphs" [32], or under the generic term "pattern" [33].

More recently, STGs form the core of one of the most commonly used frameworks for ecological successions: the *state-and-transition models (STMs)*. Note that despite their orthographic proximity, "STG" and "STM" shall not be mixed up. While STGs refer to a general mathematical concept, STMs are special instances of STGs designed for particular purposes. Indeed STMs are derived from observations and are designed to cope with the non-deterministic and irreversible nature of observed dynamics [5]. STMs are also intended to be user-friendly, enabling participatory model development and collaborative management [6]. The main goal of STMs is to assist managers and scientists in collectively proposing policies driving

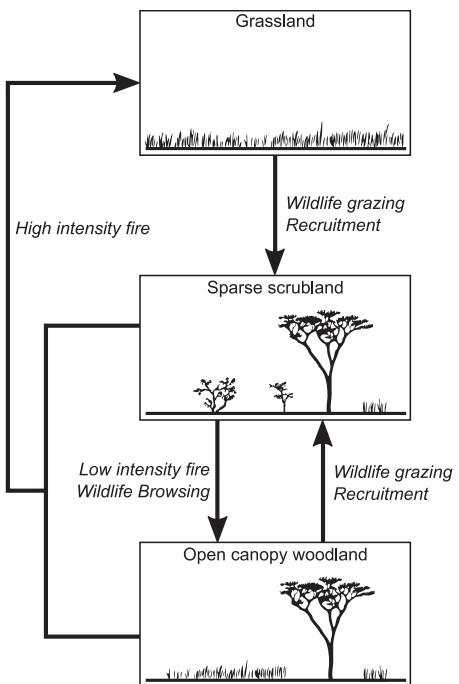
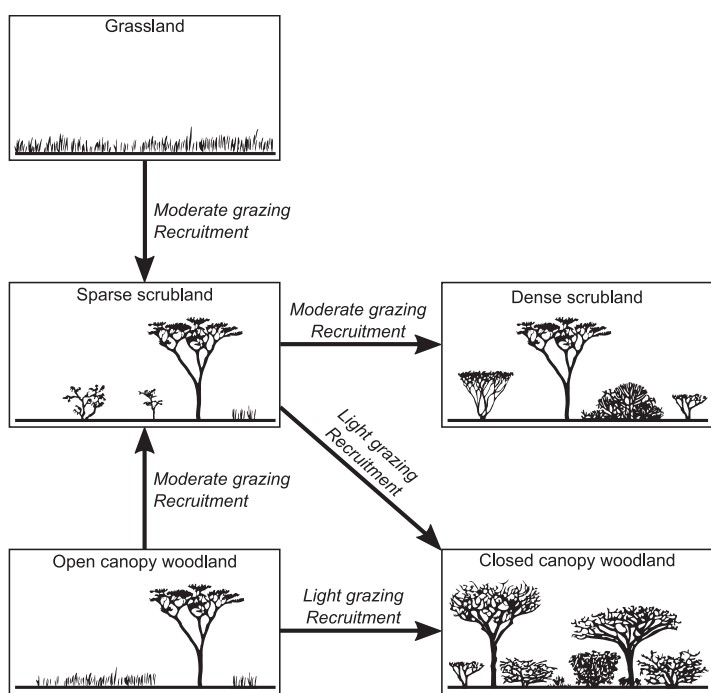

(a). Before livestock introduction

(b). With livestock and fire ban

**Fig 1. State-and-transition models of the Borana vegetation pathways.** The states are embodied by illustrated boxes, and the transitions by arrows labelled by their main driving processes. (a) Before pastoralism, fire was the main driver of the rangeland dynamics. The combination of fire, wildlife herbivory and vegetation recruitment maintained the entire system in a loop between open canopy woodland and grassland. (b) The presence of cattle and the fire ban gave a competitive advantage to woody plants, inducing an irreversible bush encroachment. Concurrently, wildlife increasingly avoided the Borana zone because of the denser human and livestock populations. (Based on [21] with author's permission).

the ecosystem through some desired pathways while avoiding others. In order to remain user-friendly, STMs sizes usually do not exceed a few dozens of states. While STMs originally stem from rangeland management [5], they are now used in many fields such as natural park management [34] (see for example the "EDIT" database housing a large catalogue of STMs [35]), geomorphology [36], or agroecology [37].

In addition, STGs are found in the field of community assembly under the concept of *assembly graphs*. In such graphs, every node is a stable species community and every edge is an invasion event. Contrarily to STMs, most studies involving assembly graphs are theoretical [7, 8], yet a few are experimental [4].

Lastly, STGs are the output of a wide diversity of modelling formalisms in ecology and environmental sciences [8, 19, 38, 39]. Recently, some studies have used Boolean models such as *Boolean networks* (i.e. systems of logical equations) to model ecological dynamics, from plant-pollinator associations [40] to spruce budworm outbreaks [41].

**1.1.2 State-and-transition models of the Borana vegetation pathways.** The STMs developed by Liao and Clark [21–24] describe the vegetation pathways of the Borana Zone in southern Ethiopia (Fig 1). Open canopy woodland (a savanna-like vegetation class encompassing a grass layer with sparse trees) was formerly the most prominent vegetation class in Borana [22]. But since the fire ban in the 1970s, the region has been undergoing a rapid increase in the density of woody plants (known as *bush encroachment*). As local people predominantly practice pastoralism, the reduction in herbaceous cover threatens their livelihood. Hence

understanding the vegetation pathways is critical to help pastoralists and policymakers mitigate bush encroachment [23].

The states of the STMs represent vegetation classes (Fig 1), see [22] for their exhaustive definitions. The transitions are labelled by their main drivers, as is often the case in the STM framework. As these STMs consist of nodes and edges representing the vegetation dynamics, they indeed form STGs. The STGs of Fig 1 are non-deterministic because some states have more than one outgoing transition. Moreover, the STG representing bush encroachment (Fig 1b) is called *irreversible* because some pathways are one-way only: for example grasslands cannot be reached from any encroached state (dense scrubland or closed canopy woodland).

**1.1.3 Building a model of Borana vegetation pathways with if-then rules.**   Most STGs found in ecology are directly drawn from observations (for example the two STMs of Fig 1). We propose to illustrate the model-checking methodology on an STG generated from a model. Based on the literature [21–24], we built a description of the Borana vegetation pathways, called "*Borana model*" in the following for concision, from which an STG can be computed. Our goal is twofold: first to show that a complex STG can be computed from a compact system description, enabling the construction of models not only based on past observations but also forecasting novel behaviours [9]; second to illustrate the scalability of the model-checking toolbox. Indeed while each STM of Fig 1 represents an observed scenario, the *Borana model* embraces the same historical scenarios as well as recommended management scenarios in order to foresee their cascading effects.

Each state of the *Borana model* consists of a vector of Boolean variables representing the functional presence (noted +) or absence (−) of the components of the system. A variable is considered functionally present if its presence has an observable influence on the system, and functionally absent otherwise. Variables influencing the system without being influenced in turn are called *controls*, for example climatic conditions or management policies. Controls remain constant along the dynamics, thus two states with distinct control values are out of reach from one another.

The transitions of the *Borana model* are generated from the execution of a rule-based formalism. More precisely, we use *if-then rules* (if the *condition* is fulfilled, then the *consequence* may arise), a methodology previously proposed [42, 43] and implemented [44] to model expert knowledge about ecosystem dynamics. Every if-then rule R whose condition is fulfilled in a given state $s \in S$ and whose consequence is not yet fulfilled in *s*, generates an outgoing transition $s \xrightarrow{R} s'$ toward the state $s' \neq s \in S$ resulting from the application of the consequence of R to *s*. Thus loops from one state to itself are excluded. Starting from a set of initial states, the full set of reachable states is computed by the cascading applications of rules. This modelling approach is exemplified by a toy model replicating the STM without encroachment (Fig 1a) and involving only three variables and four rules (Fig 2). This toy model aims for developing a fine intuition of the if-then rule modelling, note that a formal definition of if-then rule modelling is given in S1 Appendix.

The complete *Borana model* consists of 15 variables, including seven controls (Table 1), and 19 rules (S1 Table). Justifications of the modelling choices assumed by the *Borana model* are given in S2 Appendix. Each valuation of the variables describes a state of the Borana ecosystem, that can be classified into vegetation classes [22] (see S2 Table). Each valuation of the controls defines a specific *scenario* for the Borana dynamics (i.e. a combination of altitude and management policies), inspired from historical management and recommendations to limit encroachment [23]. The control variables never change in consequence of the rules (S1 Table), hence they influence the system without being influenced by it. The *Borana model* has $2^7 = 128$ initial states, one for each scenario (i.e. one for each valuation of the 7 control variables),

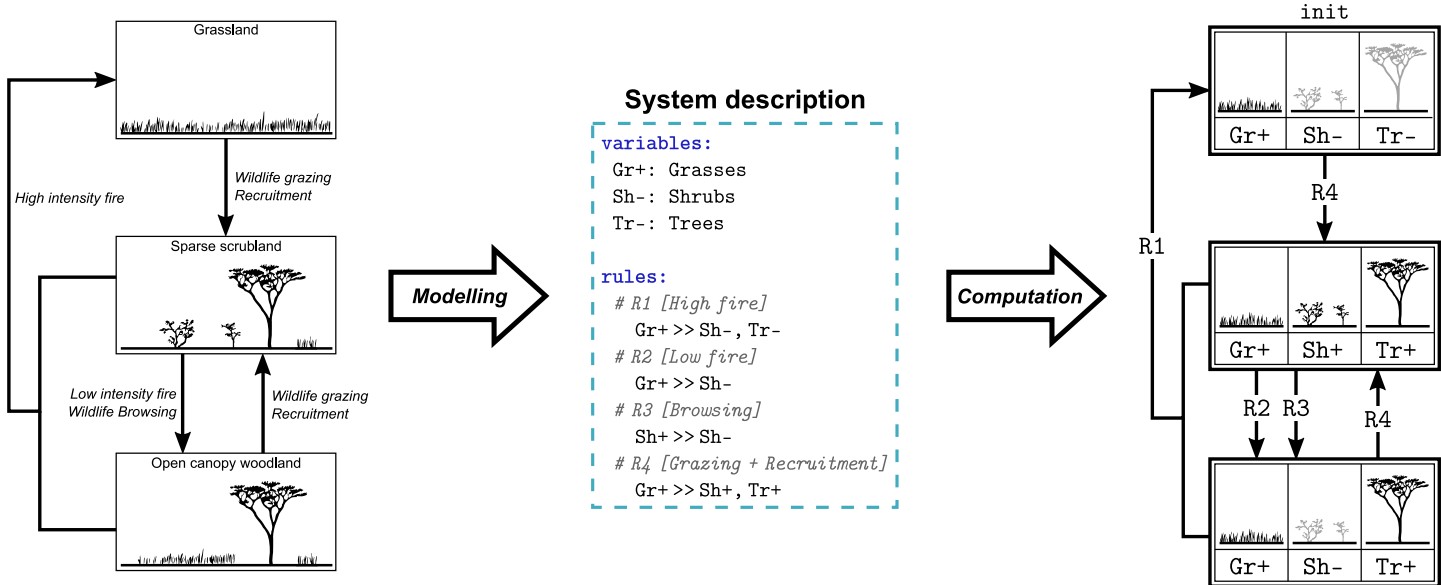

**Fig 2. Toy model illustrating the if-then rule modelling.** Modelling of the STM of Fig 1a (left) into a *if-then* rule model (middle: system description) from which an STG can be computed (right). Each state of the STG is a vector of three Boolean variables defined in the "`variables`" section of the system description: grasses (`Gr`), shrubs (`Sh`) and trees (`Tr`), noted with + if present and with − if absent. The initial values of the variables are noted next to their symbol in the system description, defining the initial state from which the STG is computed (`Gr+`,`Sh-`,`Tr-`). The "`rules`" section of the system description defines the if-then rules describing the transitions. For example, the first rule `R1` embodies that *if* grasses are present (`Gr+`) *then* (`>>`) they can fuel a high intensity fire burning down shrubs and trees (`Sh-`, `Tr-`), as grasses resprout first they do not disappear in the fire consequence. This rule corresponds in the STG to the transitions labelled by `R1` from the middle and bottom states toward the top state. The cascading applications of every rule whose condition is fulfilled and whose consequence is not to every reachable state compute the STG. Compared to the STM, the computed STG is more explicit: there are two transitions from the middle state toward the bottom one because two distinct events may lead the system from the former to the latter.

corresponding to the grassland vegetation class [21] (only grasses are present, see Table 1 and S2 Appendix). A subgraph is generated from each initial state by the cascading applications of the if-then rules. Those subgraphs are disconnected (no rule change the controls) and form together the full STG computed from the *Borana model*. Note that the toy model of Fig 2 does not have any control, thus it consists of only one connected STG.

## 1.2 Model-checking

*Model-checking* is an automated method for the analysis of any dynamical system that can be modelled by states and transitions [45]. Its goal is to check that a given automated system (hardware or software), modelled as an STG, satisfies a given dynamical property, usually

**Table 1. Variables and controls of the *Borana model*.** The initial values of the variables are noted next to their symbols.

| Variable | Description | Control | Description |
|---|---|---|---|
| `Gr+` | Grasses | `Alt` | Altitude |
| `Sh-` | Shrubs | `Fb` | Fire ban |
| `Tr-` | Trees | `Cb` | Crop ban |
| `Sa-` | Tree saplings | `Wl` | Wildlife presence |
| `Cr-` | Crops | `Ps` | Pastoralism |
| `Lv-` | Livestock | `Ig` | Intensive grazing |
| `Gz-` | Wild grazers | `BLv` | Browsing Livestock |
| `Bw-` | Wild browsers | | |

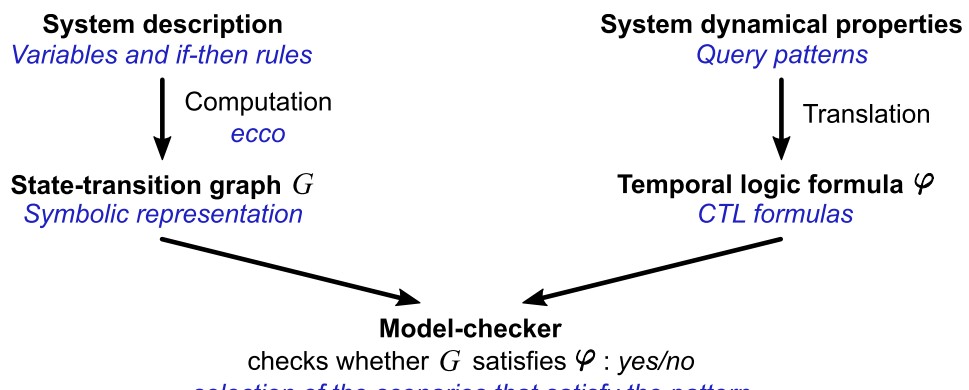

**Fig 3. The model-checking methodology.** In black the general model-checking outline. In blue italic the implementation described in this article that consists of a particular choice of techniques and tools among the available ones. (Adapted from [45]).

written as a *temporal logic formula* (Fig 3, in black). A *model-checker* is a software performing this operation, returning a *yes/no* output depending on whether the STG satisfies the property or not, generally with a counterexample pathway for negative output. In the field of computer science dealing with model-checking, STGs are mainly named *Kripke structures* or *labelled transition systems*, depending if either or both of their states and transitions are labelled. As in systems biology [17, 25], we will keep calling them state-transition graphs (STGs) for clarity. While model-checking is a wide and active field of research [2], the scope of this paper is limited to exhibiting the potential of model-checking in ecological applications. Only one implementation of the model-checking methodology will be detailed in this paper (Fig 3, blue italic annotations), while the diversity of the relevant implementations will be sketched in the Discussion.

To our knowledge, model-checking has been very scarcely used in ecology, despite its extensive application in systems biology [13]. So far, most formal analyses of STGs in ecology have been limited to graph measures [46] and topology analyses [40, 41, 44]. We identified only a few precursory applications of model-checking in ecology [19, 20]. Those studies introduce a specific implementation of the model-checking methodology based on timed automata to model the dynamics of ecosystems, such as coral reef fisheries.

Besides a modelling language that enables a description of the system that can be computed into a STG (for example the if-then rules presented above), model-checking also requires a formal language to express the dynamical properties to be checked, such as temporal logics (Fig 3).

**1.2.1 Expressing properties using Computation Tree Logic (CTL).** *Computation Tree Logic (CTL)* is one of the most popular temporal logics [2] because it is particularly fitted to express properties of branching dynamics with alternative pathways. We chose to present CTL in our implementation because ecological STGs often involves alternative pathways. A CTL formula describes a property over *computation trees*, noted *CTs* as in the beginning of CTL. A CT is rooted at a given state of the STG, and its branches are the alternative pathways starting from this state (Fig 4). In computer science, an STG represents the behaviours of a software system, thus every branch of a CT represents an alternative software computation, hence the name "computation tree". Here are two examples of CTL dynamical properties to foster intuition: (1) all the CT's pathways eventually lead to an encroached state, (2) at least one of the CT's pathways maintains grasses. A CTL model-checker checks whether the CT rooted in each

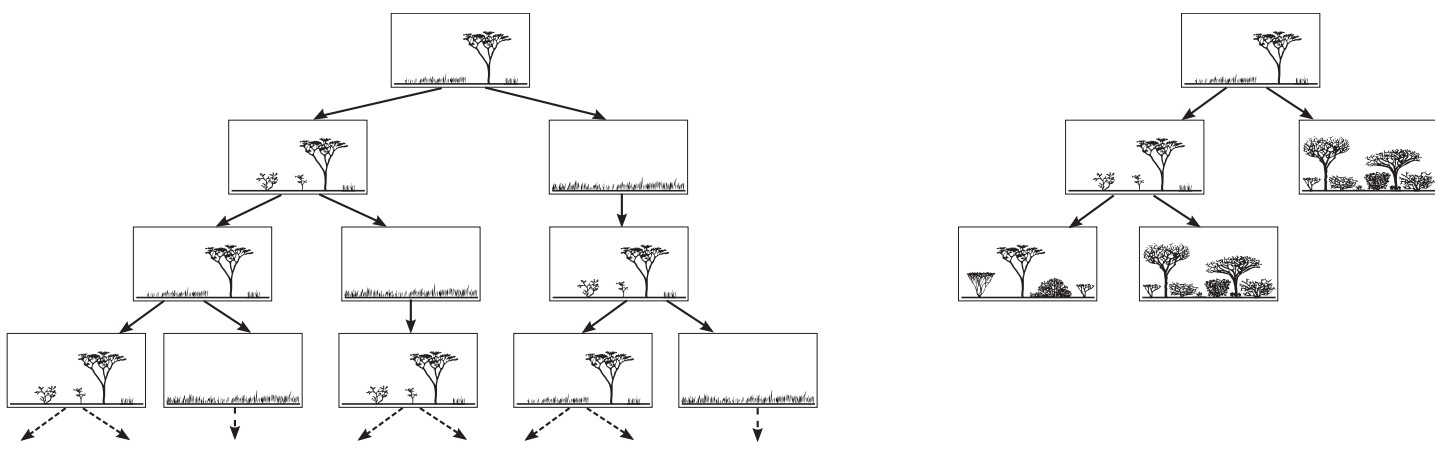

(a). Before livestock introduction

(b). With livestock and fire ban

**Fig 4. Computation trees rooted in the STMs of Fig 1.** Each branch descending from the root represents a possible pathway in the corresponding STM. (a) The CT rooted in the open canopy woodland state of the STM of Fig 1a. As the pathways are infinite in this STM, the branches of the CT are also infinite, and thus the CT itself. (b) The CT rooted in the open canopy woodland state of the STM of Fig 1b. The grassland state is not reachable from the open canopy woodland state in this STM, and thus it does not appear in its CT. As the pathways are finite in this STM, the branches of the CT are also finite, and thus the CT itself. Formally the branches of a CT are usually assumed to be infinite, so that its pathways always carry on. In order to tackle this issue, the dead-end leaves of a CT can be interpreted as infinite pathways remaining in the same state.

state satisfies a CTL formula or not. Thus a CTL model-checker discriminates between the states whose CT satisfies a given property and those whose CT does not.

The syntax and semantics of CTL are given in Fig 5. A *state property p* is a Boolean property mapping over states. For example, the presence of shrubs is a state property noted Sh+, and in Fig 2 it is only True (noted by ⊤) over sparse scrubland (middle state). More complex state descriptions are built by combining state properties using the *Boolean logical operators*: not (¬), and (∧), or (∨). For example, encroachment could be defined as the absence of grasses, and the presence of shrubs or trees: Gr− ∧ (Sh+ ∨ Tr+). Other Boolean logical operators can be built on top of the three ones above, such as the implication (⇒) that is defined such that *p* ⇒ *q* is equivalent to (¬*p*) ∨ *q*.

The *temporal operators* of CTL are always the combination of two types of operators: first a *quantifier* (∃ or ∀) dealing with branching by quantifying over the pathways starting from a given state, second a *modality* (*F*, *G*, or *U*) specifying the order of properties along a pathway. Temporal operators can thus be separated between existential and universal operators. *Existential operators* (∃*F*, ∃*G*, or ∃*U*, see Fig 5) specify that their modality has to be verified by *at least one branch* of the CT (thus by at least one pathway of the STG starting from its root state). *Universal operators* (∀*F*, ∀*G*, or ∀*U*, see Fig 5) specify that their modality has to be verified by *every branch* of the CT (thus by every pathway of the STG starting from its root state). Modality *F* specifies that the property *finally* becomes true at one step of the pathway. Modality *G* specifies that the property is *globally* true all along the pathway. Modality *U* specifies that the left-hand-side property remains true along the pathway *until* the right-hand-side property finally becomes true. Modality *next X* has been omitted from this paper to simplify the presentation.

For example, in the CTs rooted in the Borana STMs (Fig 4):

- the CTL formula ∃*F* Tr− specifies that a state without trees (the vegetation ones) is reachable from the root of the CT, which is satisfied in Fig 4a but not in Fig 4b;

# Computation Tree Logic (CTL)

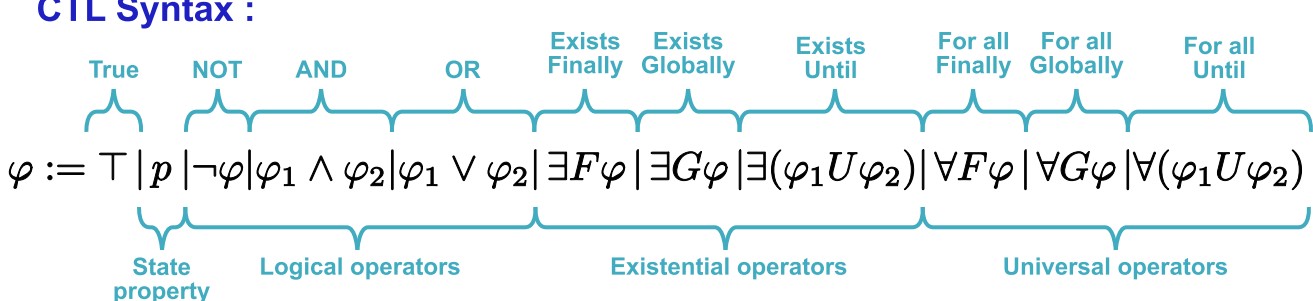

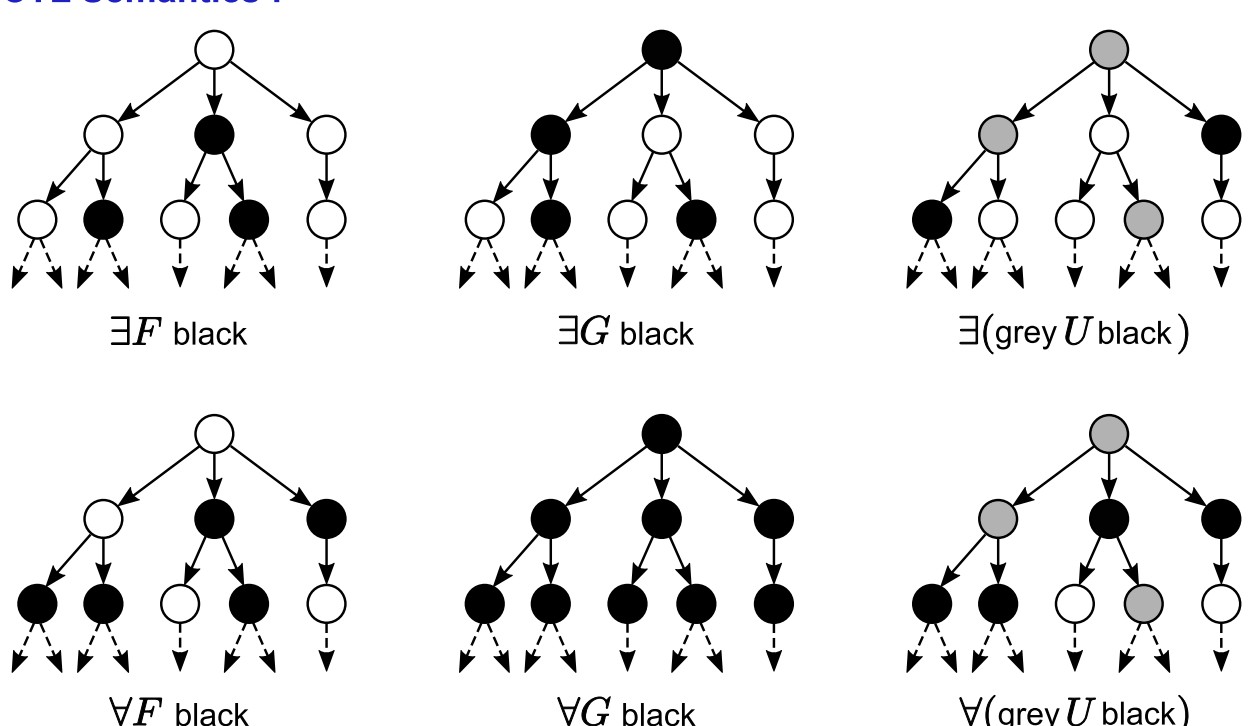

**Fig 5. Syntax and semantics of Computation Tree Logic (CTL).** The syntax defines how state properties and operators (logical, existential or universal) can be combined into a formula. The semantics describes the meaning of formulas. The semantics presented here is intuitive and given through example CTs satisfying basic CTL formulas. (Adapted from [47].) See [2] for a formal semantics of CTL.

- the CTL formula $\exists G\, \mathrm{Tr}+$ specifies that trees are always present along at least one branch of the CT, which is satisfied for the left-most branch in Fig 4a but not for its other branches, thus this CTL property is satisfied in both Fig 4a and 4b;

- the CTL formula $\forall G\, \mathrm{Tr}+$ specifies that trees are always present all along every branch of the CT, which is satisfied in Fig 4b, but not in Fig 4a.

Lastly, CTL operators can be nested to express even subtler temporal behaviour. For example, $\forall G(\exists F\, \mathrm{Tr}-)$ specifies that: all along every pathway ($\forall G$), the pathway can always branch off to reach a future state ($\exists F$) without trees ($\mathrm{Tr}-$). While $\forall G(\exists F\, \mathrm{Tr}-)$ holds in Fig 4a, the

simpler property $\forall F$ `Tr-` does not because trees never disappear in the left-most branch of the CT.

Translating a dynamical property (i.e. a description of an ecosystem behaviour) written in English into a CTL formula can turn out to be a delicate exercise for non-expert users [48]. One possible way to simplify this task is to provide users with a catalogue mapping query patterns to their translations in CTL (Table 2) [19, 49].

Note that we have illustrated the semantics of CTL by evaluating formulas with respect to a single root state and its CT. More generally, the output of a CTL model-checker is the *set* of all the states of the STG whose CT satisfies the formula. This amounts in theory to consider every state of the STG as the root of a CT, and to evaluate the formula for each CT. In practice, CTL model-checkers use much more efficient techniques to obtain this result and avoid this nested enumeration.

**1.2.2 Implementation of a scalable model-checker.**   We instantiated the general model-checking methodology from Fig 3 (in black) as indicated by the blue italic annotations, and implemented it within toolkit `ecco`. `ecco` is a Python [50] library intended to be used within Jupyter notebooks [51] and aimed at providing tools for the formal modelling and analysis of ecosystems. `ecco` has been developed and used for years to model and analyse varied ecosystems [39, 44, 52–56]. `ecco` is available as a free software released under the GNU LGPL and is hosted at http://github.com/fpom/ecco. In particular, it features an implementation of the if-then rules language presented in Section 1.1.3, as well as an efficient STG structure implemented on the top of `ITS-tools` and `libDDD` [57, 58]. At its core, `ecco` allows to compute a symbolic representation of the states of an if-then model: individual states are not explicitly enumerated, but instead a compact data structure (based on *Data Decision Diagram*, *DDD* [59]) gathers sets of states from which sets of successor (or predecessor) states can be efficiently

**Table 2. Catalogue mapping query patterns to their translations in CTL.** Dynamical properties relevant to ecological systems are gathered into patterns. The patterns are written in English and translated into CTL formulas. *x* and *y* are place-holders for state properties. (Adapted from [49]).

| English description of the pattern | CTL formula $\varphi$ |
|---|---|
| **Reachability pattern** | |
| An *x* state *can* be reached | $\exists F(x)$ |
| An *x* state *cannot* be reached | $\neg\exists F(x)$ |
| **Consequence pattern** | |
| If an *x* state is reached, then it is *possibly* followed by an *y* state | $\forall G(x \Rightarrow \exists F(y))$ |
| If an *x* state is reached, then it is *necessarily* followed by an *y* state | $\forall G(x \Rightarrow \forall F(y))$ |
| **Sequence pattern** | |
| An *y* state is reachable and is *possibly* preceded *at some time* by an *x* state | $\exists F(x \wedge \exists F(y))$ |
| An *y* state is reachable and is *possibly* preceded *all the time* by an *x* state | $\exists(xUy)$ |
| An *y* state is reachable and is *necessarily* preceded *at some time* by an *x* state | $\exists F(y) \wedge \neg\exists(\neg xUy)$ |
| An *y* state is reachable and is *necessarily* preceded *all the time* by an *x* state | $\exists F(y) \wedge \forall G(\neg x \Rightarrow \forall G(\neg y))$ |
| **Invariance pattern** | |
| *x* states *can* persist forever | $\exists G(x)$ |
| *x* states *must* persist forever | $\forall G(x)$ |
| *x* states *possibly* remain forever reachable | $\exists G(\exists F(x))$ |
| *x* states *necessarily* remain forever reachable | $\forall G(\exists F(x))$ |
| *x* states are *necessarily* reached infinitely often | $\forall G(\forall F(x))$ |
| **Reachability & Invariance pattern** | |
| It is *possible* to reach a state from which *x* states *can* persist forever | $\exists F(\exists G(x))$ |
| It is *possible* to reach a state from which *x* states *must* persist forever | $\exists F(\forall G(x))$ |

computed [60]. This symbolic approach can mitigate the *combinatorial explosion problem* (i.e. the exponential growth of the number of states with the number of variables) that is inherent to state-based approaches [61].

Our CTL model-checker, which has been integrated into `ecco`, is symbolic as well and based on the algorithm from [61]: its computes as a DDD the set of states of the STG satisfying a query formula, and a *yes/no* answer can be obtained by intersecting this set with the set of initial states. Being fully symbolic, our implementation is highly scalable and we have experimented with models up to a few billion states.

In the next Section, we use `ecco` to analyse the *Borana model* by selecting the scenarios satisfying a given CTL query, i.e. the control valuations for which the CTL formula is satisfied in the corresponding initial states. The symbolic approach enables the model-checking of a given formula at once for all the possible control valuations of the *Borana model*. By extracting only the control variables from the resulting DDD, we are able to select the scenarios satisfying a given formula. This information is then transformed into an equivalent Boolean formula that is finally transformed into a *canonical form* using SymPy [62]. This latter step has been streamlined by adding it to `ecco` as a routine directly usable on STGs.

## 2 Results

From the 128 initial states of the *Borana model*, one per scenario, an STG of 1185 states was computed by the cascading applications of the if-then rules. As controls cannot change during the dynamic, scenarios are out of reach from one another and form disconnected subgraphs of the STG. Thus every initial state spawned its own subgraph representing the dynamics of the system along the corresponding scenario and disconnected from the rest of the STG. The largest scenario subgraph has 26 states.

The subgraph corresponding to the scenario before livestock introduction at high altitude is given in Fig 6 as an example. This subgraph outputted by the *Borana model* can be compared to the corresponding STM (Fig 1a) by gathering the states belonging to the same vegetation classes (S2 Table). The subgraph and the STM are almost identical. The only difference is that in the subgraph, the transition from sparse scrubland to grassland is additionally labelled by "low intensity fire" and "browsing", because those events may happen in sparse scrubland before the establishment of trees.

The three available STMs [21] drawn from observations were compared to the subgraphs computed by the *Borana model* for the corresponding scenarios (see S1 Notebook). The first two STMs describing: (1) the pathways with wildlife herbivory and fire (Fig 1a), and (2) the pathways with extensive grazing and fire, were almost identical to their corresponding subgraphs, except for the additional labels mentioned above. The third STM describing intensive grazing with fire ban (Fig 1b), presents more differences from its corresponding subgraph. Yet the additional vegetation classes and most of the additional transitions in the subgraph were empirically observed [22]. Moreover, the subgraph showcases the main features of the STM: encroachment is not reversible, and open canopy woodland is not reachable from grassland.

We designed six CTL queries relevant to the management of the Borana ecosystem and covering all five pattern types introduced in Table 2. Those queries are built upon the following state properties:

- Closed canopy woodland is a vegetation class [22] modelled by the presence and absence of some plant variables (see S2 Table for an exhaustive definition of the Borana vegetation classes as state properties):

$$\mathrm{ClosedCanopyWoodland} = \mathrm{Gr}{-} \wedge \mathrm{Sh}{-} \wedge \mathrm{Tr}{+} \wedge \mathrm{Cr}{-}$$

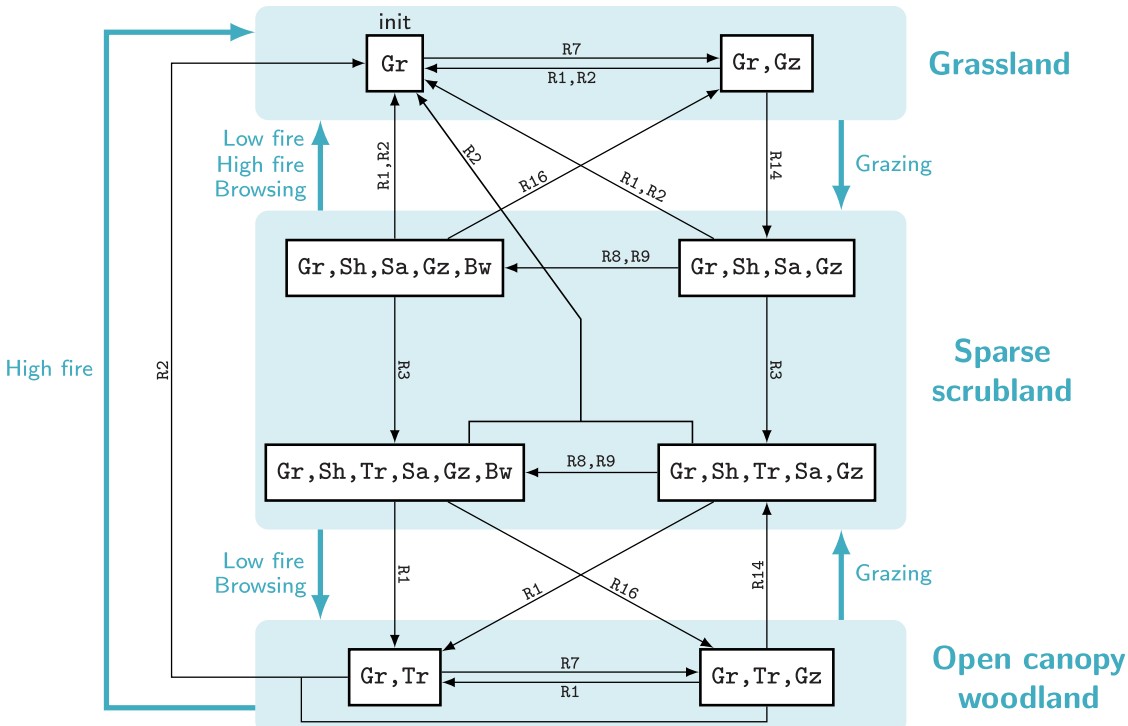

**Fig 6. Scenario subgraph computed by the *Borana model*.** This subgraph corresponds to the scenario with wildlife and fire at high altitude (`Alt+,Fb-,Cb+,Wl+,Ps-,Ig-,BLv-`), the scenario at low altitude is similar (see S1 Notebook). The states are displayed as white squares, the variables within a square represent the variables valuated "+" in this state. From the initial state (top left), the subgraph is computed by the cascading applications of the if-then rules (S1 Table). The transitions are labelled by the rules that produced them, if several rules produced the same transition then it is labelled by all of them, separated by commas. The states belonging to the same vegetation class (S2 Table) are gathered inside a blue rounded box labelled by the name of the vegetation class. Each transition from a state in a vegetation class to a state in another vegetation class, gives rise to a transition between the two classes in the same direction and labelled by the tags of the corresponding rules. For instance, transition $\{\texttt{Gr,Sh,Sa,Gz,Bw}\} \xrightarrow{\text{R1,R2}} \{\texttt{Gr}\}$ gives rise to the transition from class "Sparse scrubland" to class "Grassland" that is labelled by "low fire" (tag of `R1`) and "high fire" (tag of `R2`). The additional tag "browsing" comes from transition $\{\texttt{Gr,Sh,Sa,Gz,Bw}\} \xrightarrow{\text{R16}} \{\texttt{Gr,Gz}\}$.

- Encroachment [23] corresponds to the vegetation classes with shrubs or trees but without grass nor crop (closed canopy woodland, dense scrubland and bushland):

$$\texttt{Encroachment} = (\texttt{Sh+} \vee \texttt{Tr+}) \wedge \texttt{Gr-} \wedge \texttt{Cr-}$$

- Subsistence production [63] corresponds to the states with crops or livestock:

$$\texttt{Subsistence} = \texttt{Cr+} \vee \texttt{Lv+}$$

We used model-checking to select the control valuations (scenarios) satisfying each of the queries (Table 3). For each query and for each scenario, the model-checker tests whether the initial state of the scenario exhibits the temporal behaviour specified by the query or not, returning a *yes/no* output. We selected the valuations of the controls for which the associated model-checking output is *yes*, the omitted controls have no impact on the model-checking output. The first query exhibits a straightforward answer to the simplest pattern: encroached states are only reachable under pastoralism with intensive grazing. The second query shows that a simple pattern can have a complex answer. The remaining queries propose a general

**Table 3. Scenario selection by model-checking.** For each of the six queries we show: (1) its pattern type and CTL formula, (2) its translation into English, (3) ☑ the scenario selection (i.e. control valuations) for which the associated model-checking output of the query is *yes*, (4) an English interpretation of this scenario selection.

| | |
|---|---|
| 1) | **Reachability pattern**: $\exists F$ Encroachment <br> An encroached state can be reached. <br> ☑ Ps+ ∧ Ig+ <br> *Encroachment can only happen under the scenarios encompassing pastoralism Ps+ with intensive grazing Ig+.* |
| 2) | **Reachability pattern**: $\exists F$ ClosedCanopyWoodland <br> Closed Canopy Woodland can be reached. <br> ☑ Ps+ ∧ Ig+ ∧ (Alt+ ∨ Fb- ∨ Wl+ ∨ BLv+) <br> *Closed Canopy Woodland can only happen under pastoralism Ps+ with intensive grazing Ig+ and with at least one of the following factors: high altitude Alt+, no fire ban Fb-, presence of wildlife Wl+, browsing livestock BLv+.* |
| 3) | **Reachability + Consequence pattern**: $(\exists F$ Encroachment$) \wedge (\forall G($Encroachment $\Rightarrow \exists F\neg$Encroachment$))$ <br> An encroached state is reachable, and whenever it is reached it is possibly followed by an unencroached state. <br> ☑ Ps+ ∧ Ig+ ∧ Alt+ ∧ Cb- <br> *If an encroached state is reachable (see output Ps+ ∧ Ig+ from query 1) and if the system is at high altitude Alt+ with crops allowed Cb-, then whenever an encroached state is reached it is possibly followed by an unencroached state, i.e. encroachment is reversible.* |
| 4) | **Sequence pattern**: $\exists F($Encroachment $\wedge \exists F\neg$Encroachment$)$ <br> An unencroached state is reachable and is possibly preceded at some time by an encroached state, i.e. at least some encroachment pathways are reversible. <br> ☑ Ps+ ∧ Ig+ ∧ (BLv+ ∨ Wl+ ∨ (Alt+ ∧ Cb-)) <br> *If an encroached state is reachable (Ps+ ∧ Ig+, see query 1), there are three set of scenarios where at least some encroachment pathways are reversible: (1) with browsing livestock BLv+, (2) with wildlife Wl+, (3) at high altitude Alt+ with crops allowed Cb-.* |
| 5) | **Invariance pattern**: $\forall G(\exists F$ Subsistence$)$ <br> Subsistence states necessarily remain forever reachable. <br> ☑ (Ps+ ∧ Ig-) ∨ (Alt+ ∧ Cb- ∧ Ps+) ∨ (Alt+ ∧ Cb- ∧ Wl+) <br> *There are three sets of scenarios where subsistence remains reachable whatever happens: (1) under pastoralism Ps+ without intensive grazing Ig-, (2) at high altitude Alt+ with crops allowed Cb- and with pastoralism Ps+, or (3) at high altitude Alt+ with crops allowed Cb- and with wildlife Wl+.* |
| 6) | **Reachability & Invariance pattern**: $\exists F(\exists G$ Subsistence$)$ <br> It is possible to reach a state from which the subsistence can persist forever. <br> ☑ (Ps+ ∧ BLv+) ∨ (Alt- ∧ Ps+) ∨ (Fb+ ∧ Cb+ ∧ Ps+ ∧ Ig-) <br> *There are three sets of scenarios where it is possible to reach a state from which subsistence can persist forever: (1) under pastoralism Ps+ with browsing livestock BLv+, (2) at low altitude Alt- with pastoralism Ps+, or (3) with fire banned Fb+ as well as crops Cb+ and with pastoralism Ps+ but without intensive grazing Ig-.* |

survey of the patterns (Table 2) with answers of various complexity. Computing all the model-checking results (S1 Notebook) took only a few seconds on a modern laptop (Linux 5.4 Mint/Ubuntu, 32G RAM, CPU Intel Core i7–7820HQ 2.9GHz).

The scenario selection (Table 3) exhibits how the model-checking methodology could help better understand the Borana vegetation pathways and choose adequate management policies. The first two queries select the scenarios enabling bush encroachment. The answer to the first query shows that intensive grazing is the necessary condition for encroachment. This may seem counter-intuitive because fire has a strong influence on bush encroachment, yet bush encroachment has continued in Borana despite the lift of the fire ban in the 2000s [22, 23]. The answer to the second query shows that at least one of the following controls is additionally needed in order to reach closed canopy woodland: Alt+, Fb-, Wl+, or BLv+. Each of those controls enables one of the rules removing shrubs without changing grasses nor trees (R5, R1, R16, or R17 respectively, see S1 Table). Thus when combined with intensive grazing (R15), grasses and shrubs are removed without removing trees, resulting in closed canopy woodland.

The third and fourth queries select the scenarios making bush encroachment reversible. The third query selects the scenarios where encroachment is always reversible (from any encroached state, there is a pathway toward an unencroached state). The answer to the third query shows that crop cultivation at high altitude (Alt+ ∧ Cb- corresponding to R18, R19) is the only management policy making bush encroachment always reversible. Although this phenomenon has been observed [22], it is thought to be unfeasible at a large scale in the long term [23] due to the cost of the required inputs and the tensions between crop and

livestock agriculturists. The fourth query selects the scenarios where at least some encroachment pathways are reversible (i.e. from some encroached states, there is a pathway toward an unencroached state). The answer to the fourth query shows that in addition to crop cultivation at high altitude (Alt+ ∧ Cb−), two management policies make some encroachment pathways reversible: the presence of wildlife Wl+ and browsing livestock BLv+. Indeed pastoralists in Borana have increased their holding of browsing livestock (goats and camels) in order to mitigate bush encroachment [21, 23].

The fifth and sixth queries select the scenarios enabling subsistence. The fifth query selects the scenarios resulting in chronic subsistence (food is not constantly but only recurrently reachable). The answer to the fifth query shows that three management policies result in chronic subsistence: (1) extensive pastoralism (Ps+ ∧ Ig−), (2) pastoralism with crop cultivation at high altitude (Alt+ ∧ Cb− ∧ Ps+), and (3) crop cultivation with wildlife at high altitude (Alt+ ∧ Cb− ∧ Wl+). The first management policy corresponds to the traditional management policy in the Borana zone (nomadic pastoralism [23]), while the second policy corresponds to one of the current management policies (mixed crop-livestock systems [23]), the third management policy correspond to crop cultivation with fallow periods (which is thought to be unfeasible in the long term in drylands [23]). The sixth query selects the scenarios enabling continuous subsistence (there is a pathway along which food is constantly available). The answer to the sixth query shows that three management policies enable continuous subsistence: (1) pastoralism with browsing livestock (Ps+ ∧ BLv+), (2) pastoralism at low altitude (Alt− ∧ Ps+), and (3) extensive pastoralism without crop nor fire (Fb+ ∧ Cb+ ∧ Ps+ ∧ Ig−). This last result should be cautiously considered as continuous subsistence may be restricted to a single pathway, yet uncontrolled events may prevent humans to fully enforce this desired pathway in a real system.

The experiments on the *Borana model* illustrate the insights that model-checking can provide to ecology. Model-checking was used to validate the model by comparing its properties to empirical observations. In addition, model-checking provided a prospective analysis of the temporal behaviour of the system by foreseeing the cascading effects of management policies in order to select the ones making bush encroachment reversible or enabling subsistence.

## 3 Discussion

### 3.1 Model-checking ecological state-transition graphs

Model-checking performs efficient and automated analysis of the temporal behaviour of ecological STGs, answering a recently expressed interest in such tools [8, 9]. Given an STG and a temporal behaviour expressed as a temporal logic formula, a model-checker returns a *yes/no* output depending on whether the STG displays the behaviour or not. Model-checking is a multipurpose tool that can be used both to investigate the temporal behaviour of STGs (representing empirical data or resulting from modelling) and to validate models outputting STGs. Since model-checking is automated, it can process STGs that are too large to be examined by hand. For example, CTL model-checking was applied in systems biology to STG models made of hundreds of variables [64].

Even on the *Borana model* that consists of only 15 variables, answering questions like "*which scenarios result in chronic subsistence?*" (Table 3, query 5) would probably be hardly feasible without resorting to model-checking. Not only this question actually corresponds to a not-so-simple CTL property, but its answer is also surprisingly complex. A human examination may certainly detect the importance of pastoralism, crop ban and altitude, but would most likely fail to accurately relate them within a reasonable amount of time. On the other hand, model-checking is fully automated and provides the exact answer in a matter of seconds.

Human work is then limited to the design of temporal logic formulas, which is beneficial to scientific rigour and science reproducibility by removing ambiguity in definitions.

Yet, model-checking ecological STGs also has limitations. First, to unveil the full potential of model-checking, the size of the STG must be sufficiently large to exceed human abilities and to require an automated method. This is not always the case, especially for STMs that are often designed to be user friendly and to enable participatory practice. Yet, we believe that even in this participatory context, model-checking can still provide an adequate and rigorous conceptual framework for thinking about the dynamical properties of the STGs. Moreover, the relatively small size of most existing ecological STGs may be explained by the current lack of automated analysis tools, a lack that can be addressed with model-checking. Second, as with every automated method, the computing time of model-checking scales up with the size of its inputs (the size of the STG and the complexity of the formula). Yet the sizes of empirical STGs are limited, and studies in systems biology [16, 64] demonstrate that model-checking is able to deal with complex models. Lastly and most importantly, model-checking provides *yes/no* output (does the STG exhibits the queried behaviour or not), generally with a counterexample for negative output. The desired result may be more nuanced, for example in the *Borana model* experiments we used those *yes/no* outputs to select the scenarios where the model-checking output is *yes*. Thus the model-checking methodology may have to be slightly tweaked in order to derive more complex results from its *yes/no* output.

### 3.2 Model-checking the *Borana model*

To illustrate the model-checking of ecological STGs, we instantiated its general outline (Fig 3, in black) with a particular choice of methods and tools (Fig 3, blue italic annotations).

The system description of the *Borana model* (Fig 3, left-most half) is built upon if-then rules (Fig 2), a methodology previously proposed to model ecosystem dynamics from expert knowledge [39, 42–44]. We chose a description of the system based on events (if-then rules) because it is suited to the available data in the STM literature [6], i.e. the list of the transitions between states and their main drivers (see S2 Appendix).

In this paper, we introduced a chosen set of patterns (Table 2) and their translation into one very popular temporal logic: the Computation Tree Logic (CTL) that expresses branching properties between alternative pathways (Fig 5). We chose to represent the properties of the *Borana model* with CTL, because management actions can be represented as choices between alternative pathways. We then inputted those patterns into a model-checker in order to both validate the *Borana model* and select the scenarios achieving various management objectives (making bush encroachment reversible or enabling subsistence).

Yet this implementation of the model-checking methodology (Fig 3) is only one of the many possible implementations. For example, the system description of the *Borana model* is built upon Boolean variables, which are either present or absent. Yet in general, the model-checking methodology can be applied to any discrete-state model description computing into an STG. Thus variables can be multivalued, which is typically used in systems biology to model phenomena where a reactant regulates distinct reactions that occur at distinct thresholds. In the Borana ecosystem, fire is rare when woody plant cover is above a threshold of 40% [23, 65], thus trees could be more precisely described as multivalued rather than as Boolean: $Tr \in$ {*none*, *low*, *high*} corresponding respectively to 0%, $< 40\%$ and $\geq 40\%$. If-then rule modelling could be extended with multivalued variables, thus keeping the *Borana model* to Boolean variables was not a technical limitation but a modelling choice.

As with the system description, numerous possibilities exist to translate the dynamical properties of a system into formulas in one of the existing temporal logics (Fig 3, right-most

half). Another very popular temporal logic is the *Linear-time Temporal Logic* (LTL) [2] that expresses complex properties about a single pathway (hence its linear representation of time) by nesting temporal modalities (the same *F*, *G*, *U* we presented for CTL). For example, LTL could be used to validate models because the available observations often consist of particular pathways [66]. We did not use LTL formulas on the *Borana model* because the existing observations of the Borana ecosystem are not linear.

### 3.3 Bridges between studies and model-checking

We chose to base our overview of the model-checking methodology on the very broad STG concept [25] in order to give a uniting framework that can be specified into any particular implementation. Computer science provides with a large range of modelling formalisms computing STGs [13, 14, 67], each fitted to specific features and linked to specific model-checking software. As an exhaustive inventory of those modelling formalisms would be tedious, we limit ourselves here to sketch bridges between various existing studies in ecology handling STGs and model-checking frameworks already used in biology.

First, we emphasise that a computational model, i.e. both the system description and the computation step (Fig 3), is not mandatory in the model-checking methodology. Complex STGs can be found directly inside empirical studies [4, 68], without being computed from any underlying mathematical system description. Hence model-checking is not only a tool for the analysis of mathematical models, but can also assist the automated investigation of empirical data. In order to query such empirical STGs with a model-checking software, the STGs may be encoded into a computational model such as for example if-then rules (every transition $s \rightarrow s'$ $\in E$ would be encoded as *if s then s'*).

In ecology, STGs are often computed from interaction networks, such as differential equations [8] or Boolean networks [40, 69], with possible bridges between both [41]. Model-checking tools manipulating biological networks have been designed in the field of systems biology [67, 70], for example GINsim [71] handling Boolean networks. An example of this implementation of the model-checking methodology in systems biology is given in [17], associating a system description based on Boolean network and dynamical properties written as CTL formulas to analyse and validate a model of the mammalian cell.

When the duration of the transitions between states are available and of interest [72], they may be incorporated into the STG by labelling the transitions with their durations. Timed automata [19, 73, 74] is a modelling formalism computing such STGs, that can be implemented with the software Uppaal [75] incorporating a model-checker. A complete example of this implementation of the model-checking methodology in ecology is given in [19], associating a system description based on timed automata and dynamical properties written as Timed CTL formulas (an extension of CTL handling quantitative time) to analyse scenarios of a coral reef ecosystem.

## 4 Conclusion

This article promotes model-checking of the ecological state-transition graphs that are found in various fields of ecology, from state-and-transition models to assembly graphs. Given an STG and a temporal behaviour described as a temporal logic formula, a model-checker returns an automated *yes/no* answer to the question *"does the STG exhibit this behaviour?"*. In addition to the automated analysis of ecological STGs, we believe that definitions based on temporal logic would help clarify and compare the various concepts used in the related fields of ecology. Model-checking can be performed on both theoretical models and empirical data, as long as

they sum up into an STG. The main limitation of model-checking is its *yes/no* output, which may need to be further processed into a more nuanced answer.

Although STGs are common in ecology, the model-checking methodology remains widely unused apart from precursory studies. Yet model-checking proved in systems biology to be a valuable automated tool for the analysis of STGs, resulting into many already available software packages. The model-checking methodology encompasses a broad range of concepts and tools, thus its implementation can be fitted to the specific features of the system under study.

The main contribution of this paper is the proposition to use model-checking to assess the temporal behaviour of the various STGs found in ecology. First, we provide an inventory of ecological STGs, from historical occurrences to modern STMs and assembly graphs. Then we give a general overview of the model-checking framework, detailing the CTL temporal logic, and provide a catalogue of dynamical patterns translated into CTL. We exemplify the insights offered by model-checking through its application to a model of the Borana vegetation pathways in order to select management scenarios. Lastly, we sketch bridges between existing studies in ecology and available model-checking frameworks.

## Supporting information

**S1 Appendix. If-then rule modelling.**
(PDF)

**S2 Appendix. Justification of the *Borana model*.**
(PDF)

**S1 Table. Ruleset of the *Borana model*.** The 19 if-then rules describing the vegetation dynamics in Borana.
(PDF)

**S2 Table. Borana vegetation classes as state properties.** Borana vegetation classes [22] translated into state properties (presence or absence of vegetation variables).
(PDF)

**S1 Notebook. Python notebook covering the *Borana model* analysis.** Zip archive containing: (1) "`README`" is a text file explaining how to install `ecco` (Section 1.2.2), (2) "`Borana_model.rr`" is a text file containing the system description of the *Borana model* (Table 1 and S1 Table), (3) "`S1_notebook.ipynb`" is a Jupyter notebook covering the *Borana model* analysis (Section 2), (4) "`S1_notebook.html`" is a static HTML preview of this notebook.
(ZIP)

## Acknowledgments

We warmly thank Yann Thierry-Mieg for the fruitful discussions about the ITS-tools framework, as well as Chuan Liao for his kind permission to redraw his STMs.

## Author Contributions

**Conceptualization:** Colin Thomas, Maximilien Cosme, Cédric Gaucherel, Franck Pommereau.

**Formal analysis:** Colin Thomas, Maximilien Cosme.

**Funding acquisition:** Cédric Gaucherel, Franck Pommereau.

**Investigation:** Colin Thomas, Maximilien Cosme.

**Methodology:** Colin Thomas, Maximilien Cosme, Cédric Gaucherel, Franck Pommereau.

**Project administration:** Cédric Gaucherel.

**Resources:** Cédric Gaucherel.

**Software:** Colin Thomas, Franck Pommereau.

**Supervision:** Cédric Gaucherel, Franck Pommereau.

**Validation:** Colin Thomas, Maximilien Cosme, Franck Pommereau.

**Visualization:** Colin Thomas, Franck Pommereau.

**Writing – original draft:** Colin Thomas.

**Writing – review & editing:** Maximilien Cosme, Cédric Gaucherel, Franck Pommereau.

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
