## [Decision Letter · Decision Letter 0]

8 Feb 2022

Dear Mr. Thomas,

Thank you very much for submitting your manuscript "Model-checking ecological state-transition graphs" for consideration at PLOS Computational Biology.

As with all papers reviewed by the journal, your manuscript was reviewed by members of the editorial board and by several independent reviewers. In light of the reviews (below this email), we would like to invite the resubmission of a significantly-revised version that takes into account the reviewers' comments.

In this manuscript, the authors outline a methodological approach for studying the dynamics of discrete systems. They present the method in the context of ecosystem state transitions, e.g., recovery from a perturbation event like a fire. Given the longstanding interest in these processes, I suspect the approach will be of interest to the field and the interdisciplinary nature of the work is a strength. However, I strongly agree with reviewer 2 that the work does not fairly characterize its novelty nor its distinct contributions from past efforts. While I do not think novelty is the only criteria for a methods paper (and indeed nothing is truly novel in the strict sense) it's critical that the context surrounding the work be clearly addressed and the specific additions this method brings to the field articulated and defended with evidence. In addition, I believe the paper could be made much stronger by considering a specific data set and/or experimental system that might be studied using this approach. Given the rise in "big data" for ecosystems, demonstrating the utility of this method for such data would be of broad interest.

We cannot make any decision about publication until we have seen the revised manuscript and your response to the reviewers' comments. Your revised manuscript is also likely to be sent to reviewers for further evaluation.

Sincerely,

Samuel V. Scarpino

Associate Editor

PLOS Computational Biology

James O'Dwyer

Deputy Editor

PLOS Computational Biology

In this manuscript, the authors outline a methodological approach for studying the dynamics of discrete systems. They present the method in the context of ecosystem state transitions, e.g., recovery from a perturbation event like a fire. Given the longstanding interest in these processes, I suspect the approach will be of interest to the field and the interdisciplinary nature of the work is a strength. However, I strongly agree with reviewer 2 that the work does not fairly characterize its novelty nor its distinct contributions from past efforts. While I do not think novelty is the only criteria for a methods paper (and indeed nothing is truly novel in the strict sense) it's critical that the context surrounding the work be clearly addressed and the specific additions this method brings to the field articulated and defended with evidence. In addition, I believe the paper could be made much stronger by considering a specific data set and/or experimental system that might be studied using this approach. Given the rise in "big data" for ecosystems, demonstrating the utility of this method for such data would be of broad interest.

Reviewer's Responses to Questions

**Comments to the Authors:**

Reviewer #1: The submitted manuscript promotes the model checking technique in the area of ecological systems. The article is interesting to read, however some issues are for me not clear. Below I include point-by-point my comments:

1. Line 77: what is “EDIT STMs database”? It should be revised.

2. Line 143 Fig. 3: it should be “whether” instead of “wether”.

3. Line 257: a short technical specification of the computer that has been used should be given (including CPU, RAM, etc.).

4. Line 258 Table 3: how are the control variables embedded? It is not clear from the rest of the article.

5. Line 261: “Thus the STGs drawn below (…)” -> where?

6. In general – section 2: must be extended to include more information. It is now a little bit confusing.

7. In general – section 4: please point out the contributions of the paper.

8. In general – section 4: what are the limitations of applying model checking to ecological STGs? Appropriate information should be added.

Reviewer #2: The authors propose an interesting approach of using model-checking for scenario analysis of ecological systems.

A- General remarks on the positioning of the article:

-----

Several key points regarding the positioning of this work:

1) This approach is not new and has already been published in the following articles:

C. Largouët, M.-O. Cordier, Y.-M. Bozec, Y. Zhao, G. Fontenelle. Use Of Timed Automata And Model-Checking To Explore Scenarios On Ecosystem Models, Environmental Modelling and Software, Elsevier, 2011

M.-O. Cordier, C. Largouët, Y. Zhao. Model-Checking an Ecosystem Model for Decision-Aid. IEEE 26th International Conference on Tools with Artificial Intelligence, 2014

The authors claim that the above mentioned work is only applied in the field of agronomy, which is absolutely not true (since it is indeed about ecosystems as presented here) and that it was used in a "model-checking mindset" (what does that really mean?).

The above mentioned work was done in collaboration with ecologists (co-authors of the publication) using model-checking, query patterns, scenario testing and applied on real applications with validation of the results. This is exactly the approach that the authors wish to develop here.

The authors are asked to situate their own approach in the continuity of these previous works, precursors of the domain, in the introduction section, and to place this new work with respect to the state of the art which is not inexistent contrary to what is stated.

The authors are also asked to clearly specify where the novelty of this work lies in relation to what has already been published (which is currently unconvincing in the actual version of this paper).

2) The approach proposed by the authors is ambiguous. Several times, they defend themselves from using a "computer science" point of view and exempt themselves from giving any formal definitions. However, several concepts are difficult to grasp without more information and globally the proposed approach appears to be too vague and not enough scientifically sound.

Moreover, the aspects related to the application are not sufficiently developed in the results part. The application is only inspired by the study of a previous publication, is described as a "proof of concept" and does not propose any validation.

I therefore question the applicative side claimed by this submission.

B- General remarks on the methodology:

------

1) Choice of the model representation formalism:

Can you please give a formal definition of the STG formalism? Explain if you have defined yourself this formalism, if so how does it differ from existing approaches and what is its interest? If not, give the author's reference. In the source code, this STG seems to be a simple graph, explain if this is the case? Is it a discrete event system formalism? The definition of events is unclear. The non-determinism (L69) stated does not seem to be correct since the events on the arcs are not the same, then it is not a real case of non-determinism as usually defined.

2) Choice of the passage by the formalism of the rules:

On the method, why is it necessary to go through a rule-based system if we already have the functionning graph (Figure 2 left) as input?

- As for the STG model, the formalism of the rule-based system is not given. The article must be self-content without having to refer to another publication. It is not a usual production rule-based system which is disturbing. Otherwise, rule R1 could be triggered directly from state 1 (Gr+, Sh-, Tr-) since the premise Gr+ is consistent and is the only condition for R1 to be invoked. The R2 rule subsumes the R1 rule. How is this handled? From a practical point of view, it seems quite surprising that an expert can state both rule R1 and rule R2 (considering that we don't have the left graph). Please develop all these points to emphasize the interest of using rules in your methodology.

- How are the STG states created from the graph? I don't see how it is possible to create the STG states from these 4 rules without knowing them a priori. The logic of this step must absolutely be explained.

- In conclusion on this part, it seems to me that this rule-based system is more a graph description language than a real functioning description language.

Please explain this aspect more clearly and position yourself in relation to domain-specific languages DSL.

Reference: Fowler M, Parsons R. Domain-specific languages. Upper Saddle River, NJ: Addison-Wesley; 2011.

3) On model-checking:

- If the initial modeling language is a rule-based system why using model-checking approaches and not SAT ones that are relying on logic? Justify the choice of this approach?

Explain what is a "computation tree" even though you are in a symbolic representation (not much explained except in the implementation part when this choice appears - I suppose - as the real choice of your method).

C- General remarks on the results:

-----

This results part is very poor and must absolutely be completed if possible with a real application and at least a validation.

In particular in this part a description of the model is missing. The interest of having a STG is to be able to show a part of the provided model. It is impossible to understand what the model looks like which is very annoying for a paper that proposes to focus on ecological modeling. The "proof of concept" is not sufficient.

- The part related to the obtained model should be better presented.

It would be interesting to have a figure presenting and explaining a part of the obtained model

Where do the 1141 states come from (we simply understood that there were 8 variables and 7 controls) and why are the graphs disconnected?

- "Being fully symbolic, our implementation is 236 highly scalable and can handle models with millions of states." Did you perform scalability analysis before wrtiing such a statement? There is a need for more scientific support.

- Can the answers of model-checking queries be compared with the results obtained by Liao (type of queries, results, response time, etc.)? How can you be sure that the yes/no results are correct? The validation of the approach needs to be investigated.

D- General remarks on the discussion:

-----

The discussion, like the paper, but even more in this section, does not know how to position itself towards its readers. It proposes many concepts that are too difficult to grasp for an audience that is not in the computer science field. For more informed people, it is far too vague and not scientifically sound enough. Please find some examples here that should be explicited :

- What is management policies defined as controls? And what is it "without control"? This point should have been presented before and defined deeper in the methodology part if considered of paramount importance for the authors.

- No, probabilities do not make a system hybrid (and still questionnable for time).

- "Thus they offer fewer analysis capabilities and fewer software tools exist to support them. In general, the more expressive a particular setting is, the less it can be analyzed (because of computational complexity and tools availability), and some questions even become undecidable (i.e. no algorithm can answer them)." Moderate these assertions or support them with references.

Some problems are also undecidable with CTL so the argument is difficult to hold.

This section should be reworked in a more scientific manner.

**Have the authors made all data and (if applicable) computational code underlying the findings in their manuscript fully available?**

Reviewer #1: Yes

Reviewer #2: Yes

PLOS authors have the option to publish the peer review history of their article (what does this mean?). If published, this will include your full peer review and any attached files.

Reviewer #1: No

Reviewer #2: No
---

## [Decision Letter · Decision Letter 1]

8 May 2022

Dear Mr. Thomas,

We are pleased to inform you that your manuscript 'Model-checking ecological state-transition graphs' has been provisionally accepted for publication in PLOS Computational Biology.

Best regards,

Samuel V. Scarpino

Associate Editor

PLOS Computational Biology

James O'Dwyer

Deputy Editor

PLOS Computational Biology

Reviewer's Responses to Questions

**Comments to the Authors:**

Reviewer #1: This is a revised version of the manuscript that I have reviewed before. I appreciate the efforts of the authors to improve the quality of the article. I am satisfied with responses to my previous review as well as with changes in the manuscript.

**Have the authors made all data and (if applicable) computational code underlying the findings in their manuscript fully available?**

Reviewer #1: Yes

PLOS authors have the option to publish the peer review history of their article (what does this mean?). If published, this will include your full peer review and any attached files.

Reviewer #1: No

---

## [Editor Report · Acceptance letter]

31 May 2022

PCOMPBIOL-D-21-02081R1 

Model-checking ecological state-transition graphs

Dear Dr Thomas,

I am pleased to inform you that your manuscript has been formally accepted for publication in PLOS Computational Biology. Your manuscript is now with our production department and you will be notified of the publication date in due course.

With kind regards,

Anita Estes
